# Effects of LED Red and Blue Light Component on Growth and Photosynthetic Characteristics of Coriander in Plant Factory

Qi Gao [1,2], Qiuhong Liao [2], Qingming Li [2], Qichang Yang [2], Fang Wang [2,*] and Jianming Li [1,*]

[1] College of Horticulture, Northwest Agriculture & Forest University, Yangling 712100, China
[2] Institute of Urban Agriculture, Chinese Academy of Agricultural Sciences, Chengdu 610213, China
* Correspondence: wangfang08@caas.cn (F.W.); lijianming66@163.com (J.L.)

**Abstract:** Coriander is a whole-plant edible micro vegetable frequently used in the food industry. Its fresh eating features give it a flavor that is both tasty and refreshing, as well as potentially dangerous due to the bacteria (e.g., *Shigella sonnei*) it may contain. Artificial light-based plant factories are becoming increasingly popular due to the development of light-emitting diodes (i.e., LEDs). These plant factories employ artificial light to recreate the ideal lighting conditions for photosynthesis, ensuring plant yield and safety. Red (R) light and blue (B) light are essential for crop development and photosynthesis because R light and B light correspond to the wavelength absorption peaks of chlorophyll. However, the sensitivity of various crops to the light of varying wavelengths varies. Here, we determined the ideal R to B light ratio for cultivating coriander in plant factories by evaluating the photosynthetic characteristics of coriander ('Sumai') under different red–blue ratios. Specifically, we used monochrome red (R) and blue (B) light as controls and evaluated a total of seven different ratio treatments of R and B light (R, R:B = 5:1 (R5B1), R:B = 3:1 (R3B1), R:B = 1:1 (R1B1), R:B = 1:3 (R1B3), R:B = 1:5 (R1B5), B) under the background of uniform light intensity (200 ± 10 μmol m$^{-2}$ s$^{-1}$) and photoperiod (16-h/8-h light/dark). The results showed that the total yield of R:B = 3:1 (R3B1) was 16.11% and 30.61% higher than monochrome R and B treatments, respectively, the photosynthetic rate ($P_n$) and stomatal density were increased, and the nitrate content was decreased. Monochromatic light has adverse effects on crops. Monochromatic R light reduces the $CO_2$ assimilation amount. Monochromatic blue light treatment lowers chlorophyll concentration and net photosynthetic rate.

**Keywords:** coriander; controlled environment agriculture; artificial lighting; photosynthetic characteristics; stomatal development



## 1. Introduction

Coriander is a micro vegetable with a unique, fresh flavor; the entire plant is edible. Due to its abundance of dietary fiber, vitamin C, carotenoids, and other nutritious components [1], coriander is commonly used in the culinary industry; the stems and leaves are often used in cold meals, and the roots are used to heat broth [2]. Although coriander is popular among consumers, consuming fresh coriander poses various food safety hazards. According to studies conducted by the Federal Institute for Risk Assessment, salads and fresh spices are carriers of chemical (mycotoxins) and microbiological (pathogenic bacteria such as Salmonella) contaminants. However, the underlying source of these contaminants remains unclear. In April 2018, fresh coriander carrying *Shigella sonnei* caused illness in 33 people in the United Kingdom. Through epidemiological data analysis, this pathogen transmission was linked to the cultivation, distribution, or preparation of fresh coriander that failed to meet health safety standards [3]. Coriander is typically grown in the open air by small farming operations, which increases its susceptibility to harmful bacteria; coriander cultivation thus might struggle to meet the food safety requirements of consumers. Light-emitting diodes (LEDs) in plant factories enable crops to be produced in a fully enclosed environment under artificial control [4], which reduces pests and disease,

precludes the need to apply pesticides, and ensures the safety of crops. Therefore, incorporating coriander into plant factories can enhance its yield and food safety and increase its industrial output.

Light is a vital resource for plants that affects their morphogenesis, development, and accumulation of bioactive chemicals [5]. The intensity, quality, and distribution of light, as well as the photoperiod, regulate the light environment, and light quality has highly complex effects on plant physiology [6–8]. Light quality affects the expression of genes involved in growth and development by activating signal cascades of photoreceptors, such as phytochromes, cryptochromes, and phototropins [9,10]. Blue and red light are the most dominant and efficient wavelengths because the chlorophyll absorption spectrum in plants is most substantial in two regions: the red part of the spectrum at about 660 nm and the blue stake at about 450 nm [11,12]. Phytochromes primarily absorb R light, which plays a vital role in developing photosynthetic organs [13]. R light has the highest quantum yield of $CO_2$ fixation [14,15]. It is associated with increases in the production of rubisco, the net photosynthetic rate ($P_n$), which promotes the aboveground development of plants, mainly stem elongation [14–16]. Phototropism, photomorphogenesis, and stomatal opening are triggered by B light, primarily absorbed by cryptochromes and phytochromes [17]. Monochromatic B light is more effective than monochromatic R light in promoting the development of stomatal openings [18]. The intercellular $CO_2$ concentration ($C_i$), photosystem II primary light energy conversion efficiency ($F_v/F_m$), and photosynthetic electron transport rate (ETR) are enhanced in tomato plants under monochromatic B light [19].

Monochromatic light can have deleterious effects on plants. For example, monochromatic R light can decrease plant biomass, leaf area [20–22], leaf number [23], chlorophyll content [24], and stem elongation. The adverse effects of monochromatic R light are referred to as "R light syndrome". R light syndrome has two main causes: the initial activation of phytochrome in the absence of far-red (FR) light and the mismatch between the R and B ratios, which reduces the performance of plants [25]. Under a monochromatic B light source, plants have fewer leaves, smaller leaves and roots, and larger leaf angles with increased carotenoid content. On the one hand, this might have originated from the decreased phytochrome stimulation in the B light spectrum, which causes an imbalance between cryptochrome and phytochrome. On the other hand, due to the higher energy of blue photons compared to red ones, this can cause increased energy excess, producing photoprotective symptoms [26,27]. Therefore, a certain percentage of B light is often added to R LEDs to counteract the detrimental effects of using R or B light in isolation; this promotes a balance between cryptochrome and phytochrome and stimulates plant growth and development [28]. According to Nguyen et al. [29], a high ratio of R light, such as R:B (87:13) or R:B:FR (81.5:12.5:6), can significantly accelerate photosynthesis, promote the production of carbohydrates, and increase the biomass of coriander plants compared to monochromatic R, B, and green (G) light. Li et al. [30], Klein et al. [31], and Naznin et al. [32] also found that plants exposed to mixed R and B light exhibit high levels of Chlorophyll *a* (Chl *a*), Chlorophyll *b* (Chl *b*), total chlorophyll (Chl), and ETR, as well as the early development of non-photochemical quenching (NPQ), which enhances photosynthetic efficiency. Caferri et al. [33] found that the photosynthetic rate and stomatal conductance of basil are higher under mixed light (R:B (75:25), R:G:B (60:20:20), and R:G:B (31:42:27)) than under monochromatic R light. Zhang et al. [34] found that an R:B ratio of 7:3 increased the growth of *Salvia miltiorrhiza* Bunge as well as the content of rosmarinic acid and salvianolic acid B compared with monochromatic R and B light. Pennisi G et al. [35] explored the effects of different red and blue light ratios on basil growth and found that R:B (3:1) LED lighting improved basil yield and chlorophyll content. Spalholz H et al. [36] evaluated the effects of seven different spectral treatments on lettuce growth. They showed that the 17 d plant dry weight of lettuce grown under R:B (80:20) increased by 15–39% compared to monochromator blue light and simulated sunlight treatments. Liang et al. [37] evaluated the response of cucumber and tomato to various R and B light ratios and found that the aboveground dry weight and leaf area of cucumber were highest in the monochromatic B

light treatment; however, tomato was highest in the R:B (75:25) treatment. $P_n$, the actual photochemical efficiency of PSII ($\Phi_{PSII}$), and stomatal conductance ($g_s$) were significantly lower under monochromatic R light than under B light (R:B (50:50) and monochromatic B light) at the same light intensity as cucumber. However, no significant differences in these variables were observed among tomato treatments. Mixed R and B light can promote crop growth compared to monochromatic light, and species vary in response to the ratio of R and B light. Additional research is needed to identify the optimal ratios of R and B light for coriander cultivation.

Light quality significantly affects crop growth and photosynthetic capacity and there is a difference in the response of different crops to the spectrum. Although previous studies have evaluated that red light stimulated the yield of coriander [38,39], studies on photosynthetic characteristics are lacking. Therefore, the optimal ratio of red and blue light for coriander cultivation in plant factories must be further explored. This article takes monochromatic R and B light as the control. Seven different R:B ratios (R, R:B = 5:1 (R5B1), R:B = 3:1 (R3B1), R:B = 1:1 (R1B1), R:B = 1:3 (R1B3), R:B = 1:5 (R1B5), and B) were set to investigate the photosynthetic response to different ratios of red and blue. The parameters were mainly yield, gas exchange, Chl *a* fluorescence, and stomatal development to find the optimal red and blue light formula for coriander factory cultivation.

## 2. Materials and Methods

### 2.1. Cultivation Environment and Treatments

The experiment was conducted in an artificially lit plant growth chamber (Chengdu Academy of Agriculture and Forestry Sciences, Chengdu, China). The lighting equipment was red and blue LED light tubes (150 cm × 60 cm, Shenzhen CT Lighting Technology Co. Ltd., Shenzhen, China). The distance between the lamp plate and the cultivation plate was 45 cm. Before the experiment, spectrograms were generated using a spectroradiometer (Avantes, AvaSpec-ULS2048XL-EVO, Apeldoorn, Netherlands) with peak wavelengths of 450 and 660 nm for the B and R LEDs (Figure A1). In addition to monochromatic R and B treatments, other experimental treatments were as follows: R:B = 5:1 (R5B1), R:B = 3:1 (R3B1), R:B = 1:1 (R1B1), R:B = 1:3 (R1B3), and R:B = 1:5 (R1B5), as a specific proportion of red and blue light (Figure 1A). For all treatments, the photoperiod was 16-h/8-h light/dark, and the period of darkness was from 11:00 p.m. to 7:00 a.m.; the photosynthetic photon flux density (PPFD) was $200 \pm 10$ μmol m$^{-2}$ s$^{-1}$ (Li-1500, Lincoln, NE, USA) measured at a height of 5 cm above the culture plate. The total energy (Electric Power Detector, Longben Electric Co., Ltd., Yueqing, China) of each treatment is shown in the Appendix A (Table A1). Each treatment was repeated three times, with 24 plants per repeat.

### 2.2. Plant Material and Growth Conditions

The coriander seeds ('Sumai' Zhongle Agricultural Technology Co., Ltd., Kunming, China) were steeped in gauze for 15 min at 55 °C, followed by dark treatment for 7 d at 4 °C to germinate the seeds. The germinated seeds were placed into sponges of 2.5 cm × 2.5 cm × 2.5 cm and watered with 150 mL Enshi nutrient solution every day (pH $6.0 \pm 0.5$; EC $2.0 \pm 0.1$ ds m$^{-1}$). The seedling tray size was 32.5 cm × 24.5 cm × 4.5 cm (seeding density was 0.12 plants/cm$^2$) and cultivated for 18 d under 150 μmol m$^{-2}$ s$^{-1}$ PPFD (Li-1500, Lincoln, NE, USA) white light in an artificially lit plant growth chamber. The seedlings were then transplanted into hydroponic boxes (57.5 cm × 38 cm × 9 cm; each hydroponic tank cultivated 12 seedlings, with a planting density of 0.005 plants/cm$^2$), supplied with 11 L Enshi nutrient solution [40] (pH $6.0 \pm 0.5$; EC $2.0 \pm 0.1$ ds m$^{-1}$), controlled culture conditions at 24 °C, 48% relative humidity, and 400 ppm $CO_2$ (which is the same as the atmospheric $CO_2$ concentration) [41]. After 20 days of the seven light treatments, the plants were harvested (Figure 1B).

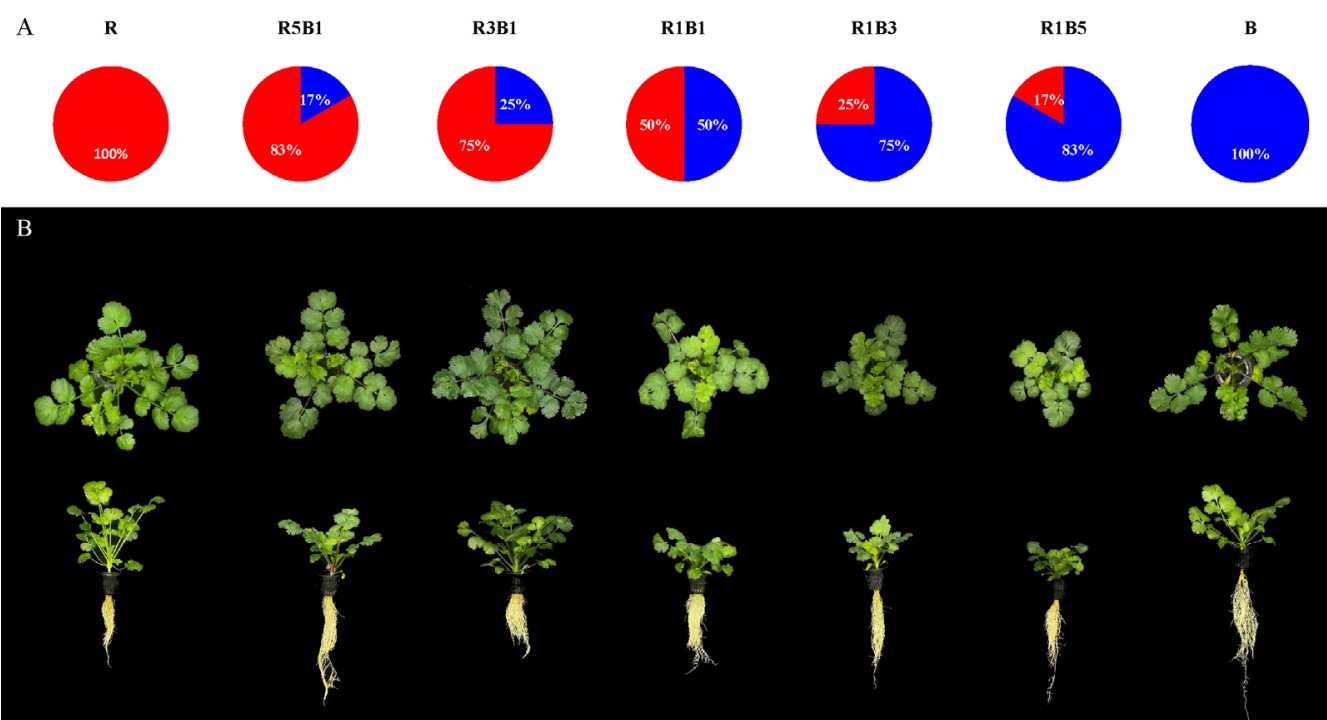

**Figure 1.** (**A**) Seven different R and B light ratio settings, respectively: R: monochrome red light treatment (200 μmol m$^{-2}$ s$^{-1}$); R5B1: red–blue ratio of 5:1 (R: 167 μmol m$^{-2}$ s$^{-1}$ and B: 33 μmol m$^{-2}$ s$^{-1}$); R3B1: red–blue ratio 3:1 (R: 150 μmol m$^{-2}$ s$^{-1}$ and B: 50 μmol m$^{-2}$ s$^{-1}$); R1B1: red–blue ratio 1:1 (R:100 μmol m$^{-2}$ s$^{-1}$ and B:100 μmol m$^{-2}$ s$^{-1}$); R1B3: red–blue ratio 1:3 (R: 50 μmol m$^{-2}$ s$^{-1}$ and B: 150 μmol m$^{-2}$ s$^{-1}$); R1B5: red and blue light ratio of 1:5 (R: 33 μmol m$^{-2}$ s$^{-1}$ and B: 167 μmol m$^{-2}$ s$^{-1}$); B: Monochrome blue light treatment (B: 200 μmol m$^{-2}$ s$^{-1}$), in which monochrome R and B light serve as control. (**B**) Top and side views of coriander plants grown under different R and B ratio treatments for 20 days (plant age from sowing: 38 days).

### 2.3. Sample Collection and Processing

After 20 days of red and blue light treatment, six plants were sampled, and their morphology, dry and fresh weight, and photosynthetic indices were determined. Then, six plants of aboveground coriander were randomly selected for the subsequent determination of physiological indexes (mixed in pairs and repeated in three groups). Before the determination, aboveground plant tissues were ground into powder in liquid nitrogen and stored at −80 °C. Finally, the top leaves of the second branch from the bottom to top of three plants were selected for chlorophyll analysis, and six plants were selected for stomatal observation.

### 2.4. Morphological Characteristics and Yield

Six plants from the same treatment were randomly selected and used for measurements of plant height, stem thickness, branch number, total leaf area, and fresh and dry weights of the roots, stems, and leaves. After taking digital photographs of each plant using a camera, the total leaf area was estimated using ImageJ 1.45 software (National Institute of Health, Bethesda, MD, USA). The plant tissue material was placed in a 70 °C electric drum bellows (DHG-2200B, Shengyuan Instrument Co., Ltd., Zhengzhou, China) to dry for 7 days before dry weight measurements were taken. The stem/leaf ratio (S/L) was determined as the ratio of stem dry biomass to leaf dry biomass [42].

### 2.5. Pigment Content

The content of chlorophyll and carotenoids was measured using the protocol of Lichtenthaler [43] with modifications. Pigment extraction was conducted using fresh coriander leaf samples (0.1 g) with 10 mL of 95% ethanol at room temperature (24 °C) until the leaves

were fully bleached. The absorbance of 200 μL of the isolated supernatant was measured at 665, 649, and 470 nm using an enzyme-labeled instrument (Flicker Biotechnology Co., Ltd., Shanghai, China). The content of chlorophylls and carotenoids was determined using the following formulas:

$$\text{Chl } a \text{ (mg g}^{-1}\text{))} = (13.95 \times A_{665} - 6.88 \times A_{649})/M \tag{1}$$

$$\text{Chl } b \text{ (mg g}^{-1}\text{)} = (24.96 \times A_{649} - 7.32 \times A_{665})/M \tag{2}$$

$$\text{Car (mg g}^{-1}\text{)} = (1000 \times A_{470}A_{649} - 2.05 \times \text{Chl } a - 114.8 \times \text{Chl } b)/245M \tag{3}$$

Formulas (1)–(3) Abbreviations: Chl *a*: chlorophyll *a*; Chl *b*: chlorophyll *b*; M: extracted mass of fresh sample/g; $A_{663.2}$: absorbance at 663.2 nm; $A_{646.8}$: absorbance at 646.8 nm; $A_{470}$: absorbance at 470 nm. The absorbance value is used in Formulas (1)–(3).

*2.6. Quality Characteristics*

The amount of soluble protein was measured following the method of Bradford [44]. Five milliliters of distilled water was mixed with 0.5 g of fresh coriander powder derived from the aboveground part of coriander plants. Then, 0.1 mL of the supernatant from the extracted solution was centrifuged at 10,000 rpm for 10 min at 4 °C and mixed with 4.9 mL of Coomassie Brilliant Blue G-250 solution (0.1 g/L; Sigma, Olympia, WA, USA). An enzyme-labeled instrument (Flicker Biotechnology Co., Ltd., Shanghai, China) was used to measure the absorbance of soluble protein at 595 nm.

The amount of soluble sugar was determined using the method of Song [45]. First, 0.5 g of crushed fresh coriander derived from the aboveground part of coriander plants was heated with 10 mL of distilled water. Then, 5 mL of sulfuric acid, 0.5 mL of ethyl anthranilate, and 1.9 mL of distilled water were added to 0.1 mL of the supernatant, which was obtained by centrifugation of the extracting solution at 10,000 rpm for 10 min at 4 °C. The absorbance of soluble sugar was measured using an enzyme-labeled instrument (Flicker Biotechnology Co., Ltd., Shanghai, China) at 630 nm.

The nitrate concentration was determined using the method of Cataldo [46]. First, 1 g of fresh coriander powder was taken and heated with 10 mL of distilled water for 30 min. Then, 0.1 mL of the supernatant was obtained by centrifugation of the extracting solution at 10,000 rpm for 10 min at 4 °C; it was then combined with 9.5 mL of NaOH (8%) and 0.4 mL each of 5% salicylic and 5% sulfuric acid. The absorbance of both solutions at 410 nm was determined using an enzyme-labeled instrument (Flicker Biotechnology Co., Ltd., Shanghai, China).

*2.7. Measurement of Physiological Characteristics*

2.7.1. Stomata

Stomata observation was made on the coriander leaves (a second branch from bottom to top parietal, L1). The method was to apply transparent nail polish on both sides of the leaf, wait for the nail polish to air dry, carefully tear off the imprint with tweezers, and then observe the dorsal axis and front pores of the leaf under an optical microscope (Olympus DP71, Olympus Inc., Tokyo, Japan) [47]. The density of the stomata was measured at 200× (nine leaves per treatment, from six plants) and the length and width of the stomata were measured at 400× (nine leaves per treatment, from six plants). Image-Pro Express software (Media Cybernetics, Silver Springs, MD, USA) was used to measure the stomata's length, width, and density.

2.7.2. Photosynthetic Characteristics

The portable photosynthetic system (LI-6800XT, LI-COR Biosciences, Lincoln, NE, USA) and R-B LED chamber (6800-01A, LI-COR Biosciences, Lincoln, NE, USA) were used to measure the photosynthetic index of coriander leaves L1. The instrument parameters were controlled as follows: temperature 24 °C, $CO_2$ concentration 400 μmol mol$^{-1}$, relative humidity 60%, flow rate 500 μmol s$^{-1}$, and leaf-to-air vapor pressure 1.0 ± 0.1 kPa. The R–B light

ratio of the photosynthetic system was matched with the R–B light ratio of the experimental treatment. Net photosynthetic rate ($P_n$), transpiration rate ($T_r$), water-based stomatal conductance ($g_{sw}$), and intercellular carbon dioxide concentration ($C_i$) were obtained. Stomatal limitation ($L_s$) was calculated as $L_s = 1 - C_i/C_a$, where $C_i$ and Ca represent intercellular and ambient $CO_2$ concentrations, respectively [48]. The maximum net photosynthetic rate ($P_{n\ max}$), fraction of light absorbed by photosystem II ($\alpha$), and dark respiration rate ($R_d$) were based on the light response model [49]. In addition, the $P_n$, $g_{sw}$ response curve was determined at 13 light intensities. The initial light intensity was 1500 μmol m$^{-2}$ s$^{-1}$ (from 200 μmol m$^{-2}$ s$^{-1}$ to 1500 μmol m$^{-2}$ s$^{-1}$ by 30 min induction), followed by 1500, 1200, 1000, 900, 800, 600, 400, 200, 150, 100, 50, 20, and 0 μmol m$^{-2}$ s$^{-1}$. $P_n$ and $g_{sw}$ measurements were logged once a steady state was achieved at each light intensity.

### 2.7.3. Chlorophyll *a* Fluorescence

Chlorophyll *a* fluorescence was measured with a portable photosynthesis instrument (LI-6800XT, LI-COR Biosciences, Lincoln, NE, USA) equipped with a fluorescence leaf chamber (6800-01A, LI-COR Biosciences, Lincoln, NE, USA). The temperature and $CO_2$ concentration were set to 24 °C and 400 μmol mol$^{-1}$. L1 leaves were dark-adapted for 2 h before measuring the parameters of chlorophyll fluorescence, and a saturating actinic light pulse of 8000 μmol m$^{-2}$ s$^{-1}$ was applied for 1 s to measure the maximal fluorescence. Data of the photosystem II primary light energy conversion efficiency ($F_v/F_m$), electron transport rate (ETR), nonphotochemical quenching (NPQ), and Initial fluorescence ($F_o$) were collected and presented. Each treatment had six bioreplicates [50].

### 2.8. Statistical Analysis

The data were processed using SPSS 17 (Chicago, IL, USA). One-way ANOVA and Duncan's multiple range test [51] were conducted to analyze significant changes among treatments, and there were significant differences when $p < 0.05$. The data were presented as mean ± standard deviation (SD). Quality index and pigment content measurement were repeated three times. All other measurement parameters were repeated six times. The experiment was repeated three times.

## 3. Results

### 3.1. Effect of the Ratios of R to B Light on Plant Morphology and Biomass

The morphology of coriander was greatly affected by the ratio of R to B light. Plant height increased with the proportion of R light, but there is little difference between B and R5B1 (Figure 2A). Stem diameter and leaf area did not differ significantly between monochrome R and B but increased significantly under R3B1 (Figure 2B,C). The number of branches was highest in the R3B1, R1B1, and monochromatic R treatments and lowest in the R1B3 treatment (Figure 2D).

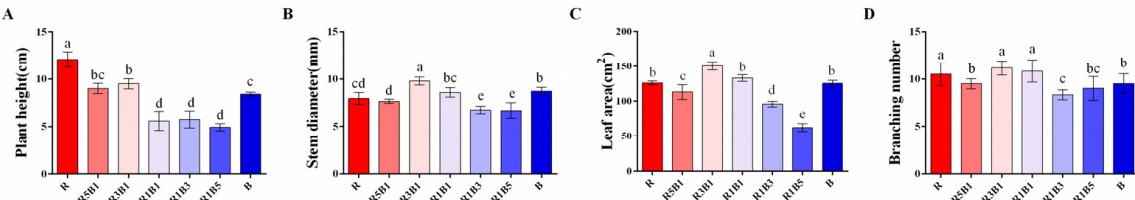

**Figure 2.** Plant height (**A**), stem diameter (**B**), leaf area (**C**), and branch number (**D**) of coriander plants as affected by the different spectra. Bars represent the mean ± SD (*n* = 6). According to one way ANOVA, different letters denote statistically significant differences between treatments (Duncan's test, *p* < 0.05). R: monochromatic red light; R1B5: red to blue ratio = 1:5; R1B3: red to blue ratio = 1:3; R1B1: red to blue ratio = 1:1; R1B3: red to blue ratio = 1:3; R1B5: red to blue ratio = 1:5; B: monochromatic blue light.

As shown in Figure 3, the dry and fresh weights of the whole plant were significantly increased under the R3B1 treatment and significantly decreased under the R1B5 treatment than those under the monochromatic R or B treatment. Specifically, the root fresh weight was 105.85% and 36.50% higher in the R3B1 treatment than in the monochromatic R and B treatments, respectively (Figure 3A). There was no difference in stem fresh weight under monochrome R and R3B1 treatment, which was significantly higher than that under monochrome B treatment (Figure 3B), and leaf fresh weight was 28.95% and 29.75% higher in the R3B1 treatment than in the monochromatic R and B treatments, respectively (Figure 3C). The yield under R3B1 treatment increased by 16.11% and 30.61% compared with the monochrome R and B, respectively. R1B5 inhibited plant growth (Figure 3D). The dry and fresh weights of the aboveground parts of the plant were similar among different treatments (Figure 3B,C,F,G). The monochromatic B treatment was advantageous in increasing stem dry weight (Figure 3B,F). Variations in the stem:leaf ratio and plant height were similar among treatments, and highest in the monochromatic R treatment, indicating that a large proportion of R light might stimulate stem growth (Figures 2A and 3H). Combined with Figures 1 and 2, overall, plants can raise yield under the R3B1 treatment, and plant morphology was optimal in this treatment. Plants in the R1B5 treatment were dwarfed and developed slowly.

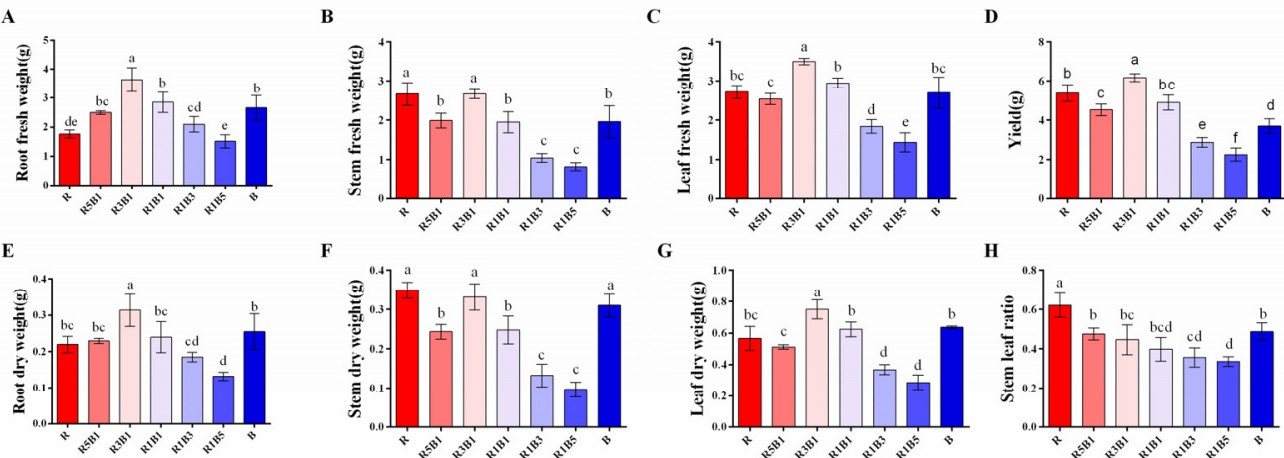

**Figure 3.** Fresh weight of each part of the plant (**A**–**C**), Yield: sum of shoot fresh weight (**D**), dry weight of each part of the plant (**E**–**G**), and stem:leaf ratio (**H**) of coriander plants as affected by the different spectra. Bars represent the mean $\pm$ SD ($n = 6$). According to one way ANOVA, different letters denote statistically significant differences between treatments (Duncan's test, $p < 0.05$). R: monochromatic red light; R1B5: red to blue ratio = 1:5; R1B3: red to blue ratio = 1:3; R1B1: red to blue ratio = 1:1; R1B3: red to blue ratio = 1:3; R1B5: red to blue ratio = 1:5; B: monochromatic blue light.

### 3.2. Effect of Various Ratios of R and B Light on Photosynthetic Pigments and Chlorophyll a Fluorescence

The ratio of R to B light had substantial effects on the pigment concentration of photosynthetic organisms. Chl *a*, Chl *b*, and the total chlorophyll content were high in the R5B1 and R3B1 treatments. The content of Chl *a* decreased significantly with the increase in the proportion of blue light, and the content was higher under monochrome R, R5B1, and R3B1 treatment with no significant differences (Figure 4A). The Chl *b* content was significantly increased with R5B1 and R3B1 treatments (Figure 4B). The total chlorophyll content was 8.88% and 7.43% higher in the R5B1 and R3B1 treatments than in the monochromatic R treatment, respectively; the total chlorophyll content was 88.37% and 85.85% higher in the R5B1 and R3B1 treatments than in the monochromatic B treatment, respectively (Figure 4B,C). There was no significant variation in the carotenoid content among treatments (Figure 4D).

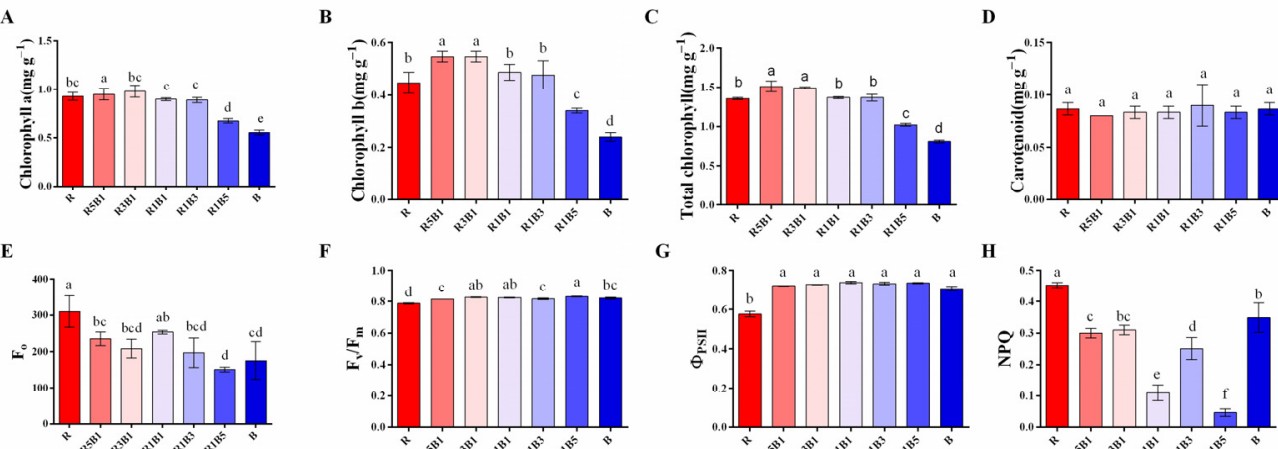

**Figure 4.** (**A–D**) represents the content of chlorophyll *a* (**A**), chlorophyll *b* (**B**), total chlorophyll (**C**), and carotenoid (**D**), respectively, under seven different spectral treatments. Bars represent the mean ± SD (*n* = 3). (**E–H**) represents the content of $F_o$: Initial fluorescence (**E**), $F_V/F_m$: PSII primary light energy conversion efficiency (**F**), $\Phi_{PSII}$: actual photochemical efficiency of PSII (**G**), and NPQ: non−photochemical quenching (**H**) under seven different spectral treatments. Bars represent the mean ± SD (*n* = 6). According to one way ANOVA, different letters denote statistically significant differences be-tween treatments (Duncan's test, *p* < 0.05). R: monochromatic red light; R1B5: red to blue ratio = 1:5; R1B3: red to blue ratio = 1:3; R1B1: red to blue ratio = 1:1; R1B3: red to blue ratio = 1:3; R1B5: red to blue ratio = 1:5; B: monochromatic blue light.

Chlorophyll *a* fluorescence was significantly affected by different proportions of red and blue light. Specifically, $F_o$ was highest in the monochromatic R and R1B1 treatments, decreasing as the proportion of R light decreased (Figure 4E). $F_v/F_m$ was significantly increased in the R1B5 treatment, and there was no significant difference in $F_v/F_m$ between the R3B1, R1B1, and R1B5 treatments. $F_v/F_m$ was 4.04% lower in the monochromatic R treatment than in the monochromatic B treatment (Figure 4F). The monochrome R treatment significantly inhibited $\Phi_{PSII}$ and there was no significant difference between the remaining treatments (Figure 4G). NPQ was highest in the monochromatic R treatment and lowest in the R1B5 treatment (Figure 4H). In conclusion, the monochromatic B light reduces chlorophyll content in plants. The Chl content and photosynthetic potential under R5B1 and R3B1 treatments were the highest compared with other treatments.

### 3.3. Effect of Various R and B Light Ratios on Photosynthetic Properties

The effects of R and B light on photosynthetic properties varied among treatments. $P_n$ was significantly increased under the R3B1 and R5B1 treatments compared to the monochromatic R and B treatments. $P_n$ decreased dramatically as the proportion of B light increased and varied little among treatments with high proportions of B light (R1B3, R1B5, and monochromatic B) (Figure 5A). $C_i$ was significantly decreased under the monochromatic R treatment and significantly increased under the R1B5 treatment compared to the monochromatic B treatment and the other mixed light treatments (Figure 5B). $T_r$ was significantly increased underR3B1 treatment compared to the monochromatic R and B treatments. $T_r$ was significantly decreased under the R1B3 treatment (Figure 5C). Compared with monochromatic red and blue light, $g_{sw}$ increased significantly under red and blue mixed light treatment (Figure 5D). $\alpha$ was significantly increased under the high proportion of red light (the monochromatic R, R5B1, R3B1, and R1B1) treatments (Figure 5E). Like the $g_{sw}$ trend of change, $R_d$ increased significantly under red and blue mixed light treatment (Figure 5F). The trend is similar to that of $g_{sw}$ and $R_d$. Compared to monochrome red and blue treatments, $P_{n\,max}$ increased significantly under red and blue mixed light treatment, except for the R1B3 treatment (Figure 5G). $L_s$ increased significantly under monochrome R light treatment and decreased significantly under R1B5 treatment (Figure 5H).

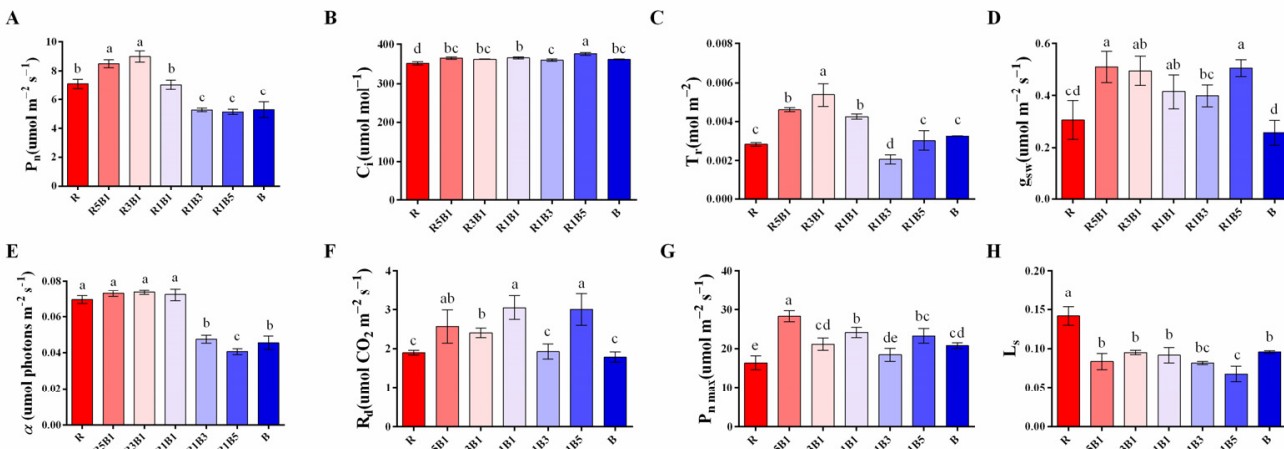

**Figure 5.** Effect of various R and B ratios on the $P_n$: net photosynthetic rate (**A**), $C_i$: intercellular carbon dioxide concentration (**B**), $T_r$: rate of transpiration (**C**), $g_{sw}$: stomatal conductance based on water (**D**), $\alpha$: fraction of light absorbed by photosystem II (**E**), $R_d$: dark respiration rate (**F**), $P_{n\,max}$: maximum net photosynthetic rate (**G**), and $L_s$: stomatal limitation (**H**) values of coriander. Bars represent the mean ± SD (*n* = 6). According to one way ANOVA, different letters denote statistically significant differences between treatments (Duncan's test, *p* < 0.05). R: monochromatic red light; R1B5: red to blue ratio = 1:5; R1B3: red to blue ratio = 1:3; R1B1: red to blue ratio = 1:1; R1B3: red to blue ratio = 1:3; R1B5: red to blue ratio = 1:5; B: monochromatic blue light.

As shown in Figure 6, $P_n$ and $g_{sw}$ increased with increasing light intensity. Except for R1B3, the light response curve increases faster under red and blue mixed light treatment compared to the monochromatic R and B light treatment. The increase in the photosynthetic rate slowed when the light intensity exceeded 600 µmol m$^{-2}$ s$^{-1}$ in the monochromatic R treatment. (Figure 6A). Compared with red and blue monochromatic light, red and blue mixed light showed higher $g_{sw}$, and the increase in the $g_{sw}$ rate slowed when the light intensity exceeded 400 µmol m$^{-2}$ s$^{-1}$ in the monochromatic R treatment (Figure 6B). Generally, photosynthesis was promoted in the mixed R and B light treatments when the R light percentage was high (monochromatic R, R5B1, and R3B1) treatments.

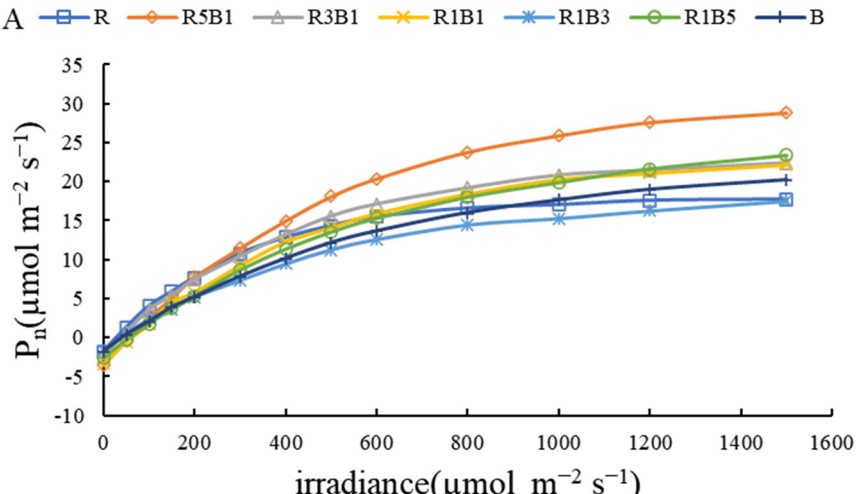

**Figure 6.** *Cont*.

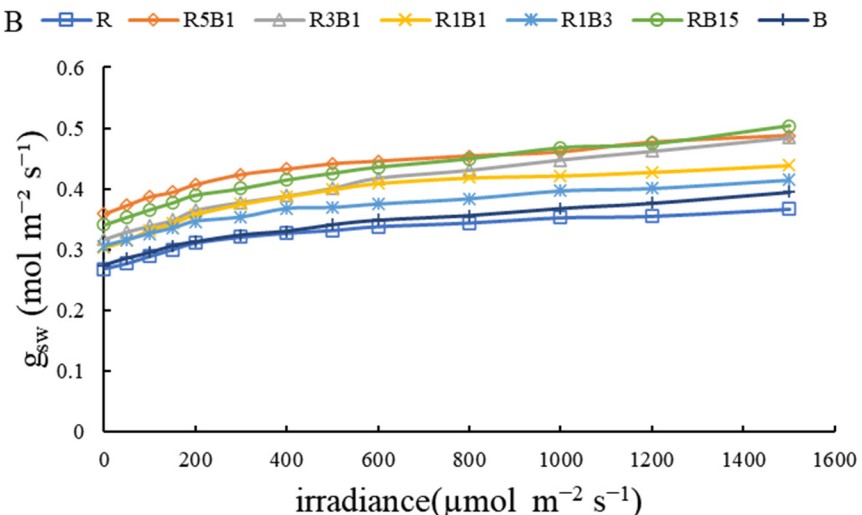

**Figure 6.** Light response curves under seven different red and blue light ratio treatments (**A**). Stomatal conductance response curves under seven different red and blue light treatment (**B**). R: monochromatic red light; R1B5: red to blue ratio = 1:5; R1B3: red to blue ratio = 1:3; R1B1: red to blue ratio = 1:1; R1B3: red to blue ratio = 1:3; R1B5: red to blue ratio = 1:5; B: monochromatic blue light.

### 3.4. Effect of Various Ratios of R and B Light on Stomatal Development

Figure 7 shows changes in the stomata on the adaxial and abaxial leaf surfaces under different R and B light ratios. Specifically, compared with monochromatic R light, R3B1 and R1B3 significantly increase the stomatal density of the adaxial and abaxial leaf surfaces. Compared to the adaxial side of the leaf, the abaxial side has a higher stomatal density (Figure 8A,E). No significant differences in stomatal density on the abaxial leaf surface were observed between the R3B1 and R1B3 treatments and the R1B1 and monochromatic B treatments (Figure 8E). The length of the stomata on the adaxial leaf surface was significantly decreased under the monochromatic B treatment (Figure 8B), while those on the abaxial surface of the leaves were the lowest under the R1B3 treatment (Figure 8F). The stomatal width of the adaxial leaf surface was lower in the monochromatic light treatments than in the mixed R and B light treatments (Figure 8C). The stomatal width on both the paraxial surface and the distal surface of the blade treated with R3B1 was significantly higher than that under the monochromatic light treatment (Figure 8G). The monochromatic B treatment significantly reduced the surface aperture width of the paraxial leaf (Figure 8D). No significant differences in aperture length on the abaxial leaf surface were observed between treatments (Figure 8F).

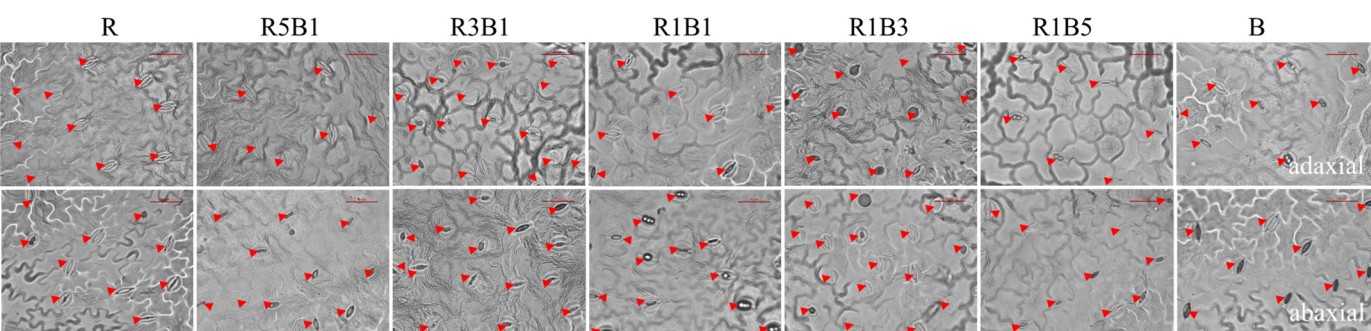

**Figure 7.** Schematic representation of the stomatal structure of the adaxial and abaxial sides of leaves under various R and B light ratios. The scale bar is 50 μm. The red triangle indicates that location of the plant stomata. R: monochromatic red light; R1B5: red to blue ratio = 1:5; R1B3: red to blue ratio = 1:3; R1B1: red to blue ratio = 1:1; R1B3: red to blue ratio = 1:3; R1B5: red to blue ratio = 1:5; B: monochromatic blue light.

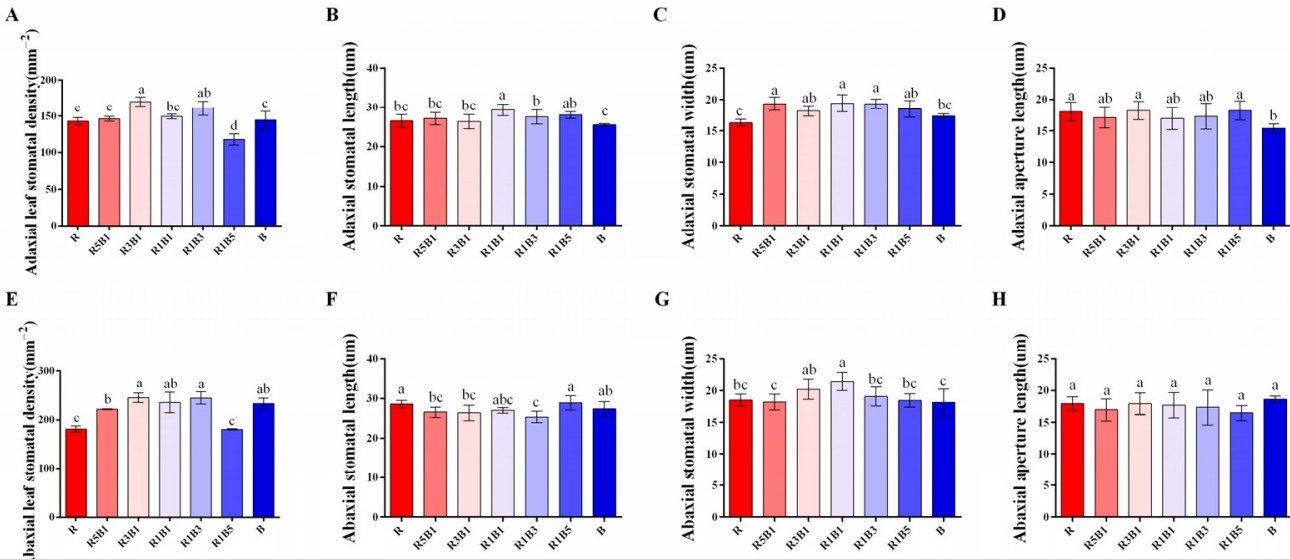

**Figure 8.** Effects of various ratios of R and B light on stomatal density (**A**,**E**), length (**B**,**F**), and width (**C**,**G**), as well as aperture length (**D**,**H**) on the adaxial and abaxial sides of leaves. Bars represent the mean ± SD (*n* = 9). According to one way ANOVA, different letters denote statistically significant differences between treatments (Duncan's test, *p* < 0.05). R: monochromatic red light; R1B5: red to blue ratio = 1:5; R1B3: red to blue ratio = 1:3; R1B1: red to blue ratio = 1:1; R1B3: red to blue ratio = 1:3; R1B5: red to blue ratio = 1:5; B: monochromatic blue light.

### 3.5. Effect of Various R and B Light Ratios on Quality

The concentration of soluble sugar was highest in the R treatment (Figure 9). The soluble sugar concentration decreased as the R light ratio decreased and was lowest in the R1B5 treatment. Variation in the content of soluble protein and soluble sugar among treatments was opposite. The soluble protein concentration increased dramatically as the proportion of B light increased; the soluble protein concentration was highest in the R1B5 treatment (Figure 9B). R3B1 and R1B1 treatments significantly reduced nitrate content (Figure 9C).

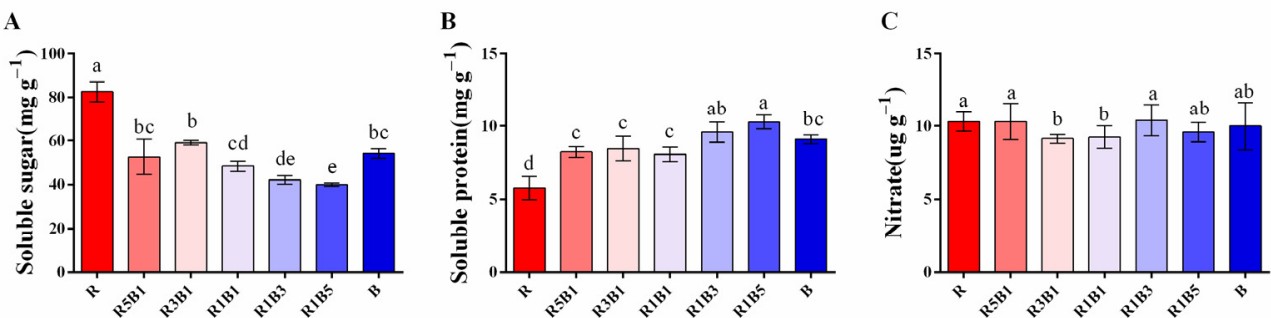

**Figure 9.** Soluble sugar (**A**), soluble protein (**B**), and nitrate levels (**C**) of coriander plants under various R and B light intensities. Bars represent the mean ± SD (*n* = 3). According to one way ANOVA, different letters denote statistically significant differences between treatments (Duncan's test, *p* < 0.05). R: monochromatic red light; R1B5: red to blue ratio = 1:5; R1B3: red to blue ratio = 1:3; R1B1: red to blue ratio = 1:1; R1B3: red to blue ratio = 1:3; R1B5: red to blue ratio = 1:5; B: monochromatic blue light.

## 4. Discussion

Spectral composition affects plants' development, growth, and morphogenesis [52,53]. Several studies have examined the effects of R and B light on plants [54,55]. Significant effects of R light on plant stem elongation have been observed [56,57] and this might have originated from the ability of R light to stimulate phytochromes, promote cell division

and expansion, and trigger hypocotyl development [58], which B light can counteract. We found that plants exposed to R light were tall and slender, but plants exposed to B light were short and robust (Figures 1B and 2). Compared with monochromatic light, mixed red and blue light promotes plant growth more [30,59,60]. When the red light ratio was high, the fresh weight, dry weight, stem length, and leaf area of cilantro plants were higher, which were most significant under the R3B1 treatment (Figures 2C and 3). Similar findings have been found in other plants, such as cucumber leaf, which area increased under the red light treatment [28]. With the increase of the R light ratio, the plant biomass of Scots Pine increased [61]. On the contrary, under a high proportion of blue light treatment, plants showed poor growth (Figure 1B), which was also shown in wheat [62] and Salvia [34]. Interestingly, the inhibition effect of monochromatic blue light on yield is not apparent (Figure 2D). Therefore, the proportion and dosage of higher blue light in indoor coriander cultivation needs further study.

Photosynthetic pigments are involved in the absorption and transport of light energy during photosynthesis [63]. Previous research has demonstrated that both B and R light treatments increase the expression of critical genes encoding chlorophyll and carotenoid pigment-related enzymes, increasing pigment accumulation, enhancing light absorption, and promoting photosynthesis [64]. In this study, the total chlorophyll concentration under monochromatic B treatment decreased significantly compared with other treatments (Figure 4C). Similar phenomena have been observed in spinach [65] and cucumber [66]. The mixed red–blue light effectively promoted chlorophyll concentration [67], confirmed again in the present study under treatment with R3B1 (Figure 4C). Light is absorbed by plants and used for photosynthesis; some light is reemitted as chlorophyll fluorescence [68]. The monochromatic red light treatment reduced $F_v/F_m$ and $\Phi_{PSII}$ and increased $F_o$ and NPQ (Figure 4E–H), indicating that the photosynthetic efficiency of chloroplasts was low and the photosynthetic system was damaged [65]. The energy of the second singlet state of Chl excited by the monochromatic blue light is almost instantaneous. The energy is then transferred to the photochemical reaction center for photosynthesis. However, adverse conditions can interfere with the photosynthetic apparatus and hinder light-driven photosynthetic electron transport, affecting light absorption [69]. It was found that monochromatic blue light reduced the total chlorophyll content and increased NPQ without a significant difference in carotenoid content (Figure 4C,D,H) to protect the photosynthetic mechanism from photo-oxidation, and finally led to the unsatisfactory utilization of incident light energy by photosynthesis [65].

The photosynthetic rate is a major determinant of plants' assimilative capacity and production. In this study, the red light treatment showed a higher net photosynthetic rate, while the blue light treatment significantly reduced the net photosynthetic rate (Figure 5A). This is due to Photosystem I (PSI) preferentially absorbing R light. Long-term exposure to R light might alter the internal balance in the excitation rate of photosystems; this might be compensated for by increases in Photosystem II (PSII), which enhances $P_n$ in the red treatment (Figure 5A) to maintain the excitation rate equilibrium of both photosystems [70]. Studies by McCree and Inada et al. also showed that the instantaneous photosynthetic rate was higher when leaves were exposed to red light [14,71]. However, with the increase in light intensity, the photosynthetic rate of monochromatic red light was significantly lower than that of other treatments (Figures 5G and 6A). On the one hand, this may be related to the PSII injury caused by the lack of blue light stimulation. On the other hand, the opposite changing trends of $C_i$ and $L_s$ under the monochromatic R light treatment indicated that the lower $g_{sw}$, along with the increase in light intensity, might be another reason for the decrease of $P_{n\,max}$ [72]. Many studies have shown that adding blue light on a red background can effectively promote crop photosynthesis [73,74]. In this study, R3B1 and R5B1 showed the maximum $P_n$, $P_{n\,max}$, and $\alpha$ (Figure 5A,E,G) because the mixed light of red and blue significantly promoted the activities of enzymes related to photosynthetic transformation and enhanced photosynthesis [75]. In contrast, the high proportion of blue light treatments decreases $P_n$ and $\alpha$ (Figure 5A,E), which may be related

to impaired mesophyll conduction and chloroplast avoidance response [76,77]. Different crops have different responses to different wavelengths [78]. In this study, monochromatic red light and high-proportion red light treatment promoted the growth of coriander. At the same time, the photosynthetic rates of rice [54] and wheat [20] were inhibited under monochromatic red light. This indicated that genetic variation might affect growth-related decreases in photosynthetic rate.

Stomata are essential for exchanging air and water with the surrounding environment. Various types of light affect the dynamic behavior of stomata, which in turn affects the photosynthetic performance of plants [79]. The responses of stomata to red and blue light were mainly divided into two parts [80,81], a component independent of photosynthesis and a component dependent on photosynthesis. Blue light applied to the former could induce rapid stomatal opening [82]. The latter is driven by red light through photosynthesis [83]. Under the monochromatic R light treatment, the stomata showed the characteristics of "more and larger" with longer pore size and lower density (Figure 8A,D,E,H), which was similar to the stomatal development of lettuce under the bottom illumination treatment, leading to the reduction of stomatal conductance [84]. $g_{sw}$ represents the ability of plants to absorb carbon through stomata, which affects the photosynthetic rate [85]. B light stimulates stomatal growth [86,87]. In this study, we found that the monochromatic B treatment increased the stomatal density on the adaxial leaf surface (Figure 8E), and similar patterns have been observed in lettuce [7,21] and chrysanthemum [88]. This is caused by the fact that B light suppresses the cop1-mediated degradation of the ICE protein, which stimulates stomatal growth [89]. Red and blue light has an additional effect on the stomatal opening [85]. This study has confirmed this theory. Compared to monochromatic light, the red–blue mixed light has a higher $g_{sw}$ (Figure 5B). This advantage persists with the increase of light intensity (Figure 6B), and this likely stemmed from the fact that mixed R and B light treatments with a high proportion of R light promoted stomatal development by upregulating the expression of stomatal development genes such as ZmSPCH and ZmMUTE [90].

## 5. Conclusions

The light spectrum of the surrounding environment has a significant impact on the growth and development of plants. In this study, monochromatic light harmed the growth of coriander. Adding a certain proportion of blue light based on red light could alleviate this negative effect. Monochromatic R light weakened $CO_2$ assimilation and decreased plants' potential maximum photosynthetic capacity; monochromatic B light treatment caused a decrease of $P_n$ and stomatal aperture length of coriander. The negative effect of monochromatic R may be caused by decreased stomatal conductance and density. The negative effect of monochromatic B might be related to the decrease in chlorophyll concentration. R3B1 light treatment under the mixed red and blue light increased the stomatal density, chlorophyll concentration, and net photosynthetic rate. It significantly increased crop yield and reduced nitrate content, which was the optimal ratio of red light to blue light for the growth of coriander. On the contrary, the high proportion of blue light treatment inhibited the growth of coriander. Interestingly, the inhibition of monochromatic blue light on yield was insignificant. Therefore, the proportion and dosage of higher blue light in indoor coriander cultivation need to be further studied. To meet the production needs of the factory, the recommended ratio of R:B is 3:1.

**Author Contributions:** Conceptualization, J.L. and F.W.; writing—original draft preparation, Q.G. and F.W.; writing—review and editing, Q.L. (Quihong Liao), Q.L. (Qingming Li), Q.Y., F.W. and Q.G.; funding acquisition, J.L. and F.W. All authors have read and agreed to the published version of the manuscript.

**Funding:** This research was funded by the Sichuan Science and Technology Program, grant No. 2022JDRC0108, the Local Financial Project of the National Agricultural Science and Technology Center, grant No. NASC2020KR01 and the Agricultural Science and Technology Innovation Program, grant No. ASTIP2022QC03, 34-IUA-03&34-IUA-01.

**Data Availability Statement:** Not applicable.

**Conflicts of Interest:** The authors declare no conflict of interest.

## Appendix A

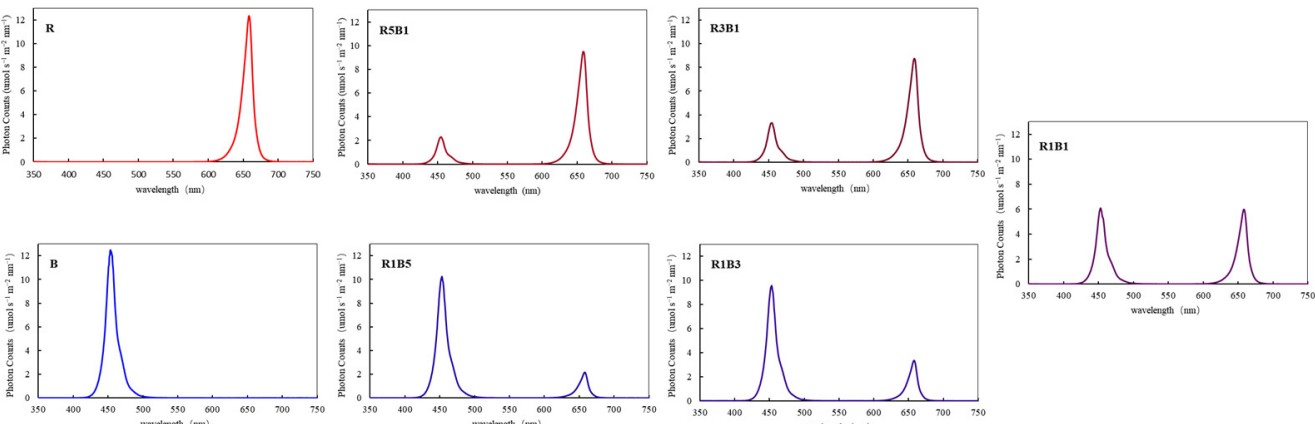

**Figure A1.** The spectra treated with different proportions of red and blue are shown in the figure. R: monochromatic red light; R1B5: red to blue ratio = 1:5; R1B3: red to blue ratio = 1:3; R1B1: red to blue ratio = 1:1; R1B3: red to blue ratio = 1:3; R1B5: red to blue ratio = 1:5; B: monochromatic blue light.

**Table A1.** The power under different proportions of red and blue treatments is shown in the table. R: monochromatic red light; R1B5: red to blue ratio = 1:5; R1B3: red to blue ratio = 1:3; R1B1: red to blue ratio = 1:1; R1B3: red to blue ratio = 1:3; R1B5: red to blue ratio = 1:5; B: monochromatic blue light.

| Treatment | Power (w s$^{-1}$) | Electricity Consumption (kw h$^{-1}$) |
|:---:|:---:|:---:|
| R | 140.5 | 2.248 |
| R5B1 | 139.2 | 2.2272 |
| R3B1 | 134.1 | 2.1456 |
| R1B1 | 129.8 | 2.0768 |
| R1B3 | 127.8 | 2.0448 |
| R1B5 | 125.4 | 2.0064 |
| B | 119.2 | 1.9072 |

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
