# Peer review of "Effects of LED Red and Blue Light Component on Growth and Photosynthetic Characteristics of Coriander in Plant Factory"

_horticulturae, doi:10.3390/horticulturae8121165_

Round 1

Reviewer 1 Report

Authors may find my comments in the attached .pdf document.

Reviewer 2 Report

Dear Authors,

the work aims at investigating coriander photosynthetic response to different ratios of red (R) and blue (B) light, including monochromatic light treatments, i.e., R100B0 and R0B100. The investigation is justified only by the need to increase food safety but also the increased morphological and nutritional quality obtained in indoor farming should be considered.

Please read below main comments and check the attached pdf for complete overview of the comments.

In addition, Abstract is too short and completely lacks info on the M&Ms.

Introduction needs more references to validate what authors are communicating and also to compare to similar studies already reported in literature.

M&M sections need more details on procedure and materials used, important info such as cultivation environment, cultivar tested, etc. Additionally, statistical methodology should be explained in a clear way, including how many times the experiment was performed and the number of replicates used for each variable as different numbers reported in various sections create confusion, e.g. n=3 in graphs, n=6 in section 2.7.

Results needs to be shortened. Authors need to report only most relevant results.

Discussion should absolutely include reference to any inference. Also, reporting more work that compare different R & B light ratios would be more beneficial than spending most of the focus on monochromatic light treatments which are already known as non optimal. Also highlight the focus on photosynthetic traits.

Conclusion should be rewritten with a different approach as suggested in the pdf file.

Figures should be improved by increasing the text size as it is currently hard to read and by adding info in captions.

Kind regards

Round 2

Reviewer 1 Report

Dear Authors,

Thank you for making the changes suggested by both reviewers to the manuscript. This has clearly improved the paper. I would suggest going over the manuscript a few more times and making minor grammatical edits (e.g., using passive voice in parts of M&M, etc.). After these changes are made, I will find the manuscript suitable for publication in the journal.

Best regards,

Author Response

Dear reviewer,

Thank you very much for your suggestion. We have read the manuscript repeatedly, simplified the lengthy sentences to clarify the paragraphs, deleted the experimental methods not mentioned in the result part, and corrected the irregular writing style. The details are in the word.

Sincerely yours,

Qi Gao

Reviewer 2 Report

Dear Authors,

Good job in addressing all comments.

Kind regards

Author Response

Dear reviewer,

Thank you very much for your approval of the revised results, and thank you again for your valuable suggestions on the article.

Sincerely yours,

Qi Gao